# Transperineal Vulvar Ultrasound: A Review of Normal and Abnormal Findings with a Proposed Standardized Methodology

**DOI:** 10.3390/diagnostics15050627

**Published:** 2025-03-05

**Authors:** Nina Montik, Camilla Grelloni, Giovanni Delli Carpini, Jessica Petrucci, Jacopo Di Giuseppe, Andrea Ciavattini

**Affiliations:** Gynecologic Section, Department of Odontostomatologic and Specialized Clinical Sciences, Università Politecnica delle Marche, 60123 Ancona, Italy; nina.montik@ospedaliriuniti.marche.it (N.M.); c.grelloni@pm.univpm.it (C.G.); giovanni.dellicarpini@ospedaliriuniti.marche.it (G.D.C.); j.petrucci@pm.univpm.it (J.P.); jacopo.digiuseppe@ospedaliriuniti.marche.it (J.D.G.)

**Keywords:** vulva, ultrasound, HFUS, ultrasonography, vulvar neoplasms, vulvar lichen sclerosus

## Abstract

The vulva is a complex anatomical organ that may present with a wide range of pathologies. Even if it can be easily investigated, correctly interpreting vulvar appearance is often challenging. Vulvar ultrasound is an emerging diagnostic technique that may be helpful in different aspects of vulvar pathology. We aimed to summarize the state of the art of vulvar ultrasound, provide the necessary theoretical bases of embryology and anatomy, describe the normal and pathological vulvar sonographic characteristics, and propose a feasible and reproducible methodology for vulvar ultrasound. Vulvar sonographic scan should be performed with a linear probe, preferably > 15 mHz, following a standardized methodology. The sonographic appearance of the normal vulva reflects the different histology of its structures and, thus, their embryogenetic origin. The description of a suspected vulvar lesion should include localization, dimensions, volume, type of growth, shape, appearance of the edges, depth of invasion, echogenicity, and identification of vascularization. Cystic dilatation of obstructed Bartolini ducts is the most common benign finding in the vulva (fluctuant structures in the posterior third of the labia majora containing clear mucous fluid). Malignant vulvar lesions appear as hypoechogenic or heterogeneous solid lesions with irregular margins and a high degree of vascularization. Extramammary Paget Disease presents a homogeneous hypoechogenic creeping area in the epidermis due to neoplastic cells typical of this disease. The potential applications of vulvar ultrasound are examining the content of a vulvar swelling to guide its management and assessing the response to medical treatment in the case of lichen sclerosus. In managing patients affected by vulvar malignancies, it may play a critical role in local staging, stromal invasion determination, measuring the distance from the midline, and assessing the eligibility for sentinel lymph node procedure. Vulvar ultrasound is a minimally invasive and economical test that can be performed with minimal equipment. Further studies will be necessary to validate the clinical applications, quantify the diagnostic performance, and evaluate the agreement between operators.

## 1. Introduction

The vulva is a complex anatomical organ composed of multiple soft tissue and skin layers that may present with a wide range of pathologies (inflammatory, autoimmune, preinvasive, or neoplastic) [1]. Due to its constitution and location, visual inspection can easily investigate the superficial vulvar appearance. If abnormal findings are noted, second-level examinations, such as vulvoscopy, may be performed with the potential need for a vulvar biopsy for the final diagnosis. However, the correct interpretation of vulvar appearance is often very challenging, with difficulties in the distinction between nonmalignant and malignant lesions or in the correct evaluation of the deep extension of a lesion. Ultrasound imaging is an inexpensive, feasible, fast-to-perform, non-invasive diagnostic tool that does not require exposure to ionizing radiation and, therefore, has an excellent safety record [2]. While transvaginal and transabdominal US use has become part of the clinical practice in gynecology and obstetrics, the ultrasonographic study of vulvar structures has not yet been fully developed. Since the first publication of the sonographic study of skin thickness in 1979, the availability of more technologically advanced ultrasound probes allowed a wider diffusion of this technique by obtaining increasingly precise and detailed ultrasonographic images [3,4]. More specifically, the introduction of high-frequency ultrasound (HFUS) probes (above 20 MHz) with aligned broadband transducers allows us to distinguish the different skin layers better and perform more accurate diagnoses of skin lesions [5]. Following the availability of HFUS, the application of transperineal vulvar ultrasound in clinical practice is assuming growing interest. The first studies on this topic provided the sonographic description of vulvar anatomy [6] and reported that HFUS might allow a better evaluation of vulvar lesions by studying vascularity, echogenicity, and definition of affected layers that cannot be evaluated by vulvoscopy [4]. Vulvar ultrasound has a wide range of potential applications. It may be helpful to distinguish between benign and malignant masses, evaluate the content of a lesion, assess the response to medical treatment [7,8], and perform the pre-operative evaluation of vulvar malignancies. In this case, ultrasound may allow us to estimate the extent of the lesion, to measure the thickness of stromal invasion, and to better evaluate the involvement of adjacent structures, like urethral and/or vaginal mucosa, bladder mucosa, or rectal mucosa [9]. Performing a vulvar ultrasound investigation requires a deep knowledge of vulvar embryology, histology, and gross anatomy to correctly interpret the normal and pathological sonographic characteristics. Therefore, we aimed to summarize the state of the art of vulvar ultrasound, provide the necessary theoretical bases of embryology and anatomy, describe the normal and pathological vulvar sonographic characteristics, and propose a feasible and reproducible methodology for vulvar ultrasound. 

## 2. Embryology

All three embryonic layers (endoderm, mesoderm, and ectoderm) contribute to forming the female external and internal genitalia [10]. During the 3rd to 4th weeks of gestation, the development of the external genitalia begins from the proliferation of the intermediate mesoderm along the posterior wall of the abdominal cavity to form the genital folds, located on both sides of the caudal end of the hindgut called the cloaca, made up of endoderm and covered by a membrane made of ectoderm [10]. Lateral to the genital folds, starting from the undifferentiated stages, the labio-scrotal swellings (also called genital swellings, ectoderm-derived) develop, constituting the future labia majora. In the most cranial portion of the cloacal membrane, the mesodermal cells of the genital folds fuse along the midline to form the genital tubercle [10]. Starting from the 5th week, the cloacal membrane is progressively divided by the urorectal septum (the future central tendon of the perineum, derived from the mesoderm) into the urogenital membrane ventrally (surrounded by the genital folds) and the anal membrane dorsally. At the end of the 9th week, the urogenital and anal membranes disappear, giving rise to the urogenital sinus and the anal canal, respectively [10]. The differentiation of the external genitalia in male or female direction is visible after the 12th week of development. In the females, due to the estrogenic and testosterone-absent environment, the growth of the genital tubercle (or phallus) will form the corpora cavernosa and the clitoris. The genital and labioscrotal folds remain largely unfused and give rise to the labia minora and majora, respectively [11]. The most cranial areas of the united genital folds form the frenulum of the clitoris. Anteriorly, the labia majora fuse to form a raised area called the mons pubis, while cranially to the anus, they fuse into the posterior labial commissure and perineal body [11]. Vaginal development starts in the 7th week and involves both the Müllerian ducts and the urogenital sinus. The caudal end of the merged Müllerian ducts forms the uterovaginal primordium that reaches the posterior part of the urogenital sinus. This process results in the formation of the sinus tubercle, which induces paired endodermal outgrowths to form sinovaginal bulbs. The sinovaginal bulbs connect the urogenital sinus and the uterovaginal primordium and then fuse to form a vaginal plate that will finally produce the inferior two-thirds of the vaginal canal [11,12]. The urogenital sinus partially degenerates to create an open lumen that extends to the vaginal entrance and expands to form the vestibule; therefore, both the vaginal epithelium and the vulvar vestibule are of endodermal origin [11]. The distal urethra and the anus also derive from the urogenital sinus, so they are of endodermal origin. Figure 1 summarizes the main phases of embryogenesis of female external genitalia, and Table 1 summarizes the embryogenic origins and precursors of the vulvar anatomical parts.

## 3. Vulvar Anatomy

The external female genitalia, or vulva, is a complex anatomical organ located inside the urogenital triangle. Its main constituents are the pubic mound, the labia majora, the labia minora, the vestibule and the clitoris [13]. The mons pubis, a pad of adipose tissue overlying the pubic symphysis, takes up the most cranial portion of the vulva, while caudally, it continues into the labia majora and posteriorly rejoins in the perineal body [13]. The genitocrural folds are the external lateral borders of the vulva [14]. Labia majora medially borders the interlabial folds, which separate them from the labia minora, representing a transition point where the skin becomes glabrous [10]. The interlabial folds identify and separate the central and lateral portions of the vulva [14]. Labia minora are two fat-free pigmented skin folds that can vary in morphology and size. They surround the vaginal orifice and delimit the opening into the vestibule. Anteriorly, they bifurcate into a medial fold that unites posterior to the clitoris to form the frenulum of the clitoris and a lateral fold that merges anterior to the clitoris to form the prepuce of the clitoris [10]. The borders of the labia minora constitute an imaginary circular line, also called Hart’s line, that demarcates the vestibular region, which is of endodermal origin, from the exterior elements of the vulva, which are of mesodermal/ectodermal origin. Therefore, Hart’s line starts anteriorly on the prepuce of the clitoris, runs laterally along the labia minora, and posteriorly reaches the vaginal fourchette [10,13]. Within the vestibule are the paraurethral vestibular glands and the vaginal and urethral meatuses. The hymen is a connective membrane that bounds the vaginal orifice with a central opening through which there is the passage of menstrual blood. The hymenal ring constitutes the internal border of the vulva [14]. The clitoris is a complex structure comprising two corpora cavernosa characterized by erectile tissue, glans, suspensory ligament, root, paired crura, and vestibular bulbs. The external portion of the clitoris, called the “glans”, is covered by the prepuce anteriorly and bordered by the frenulum posteriorly [13]. Figure 2 shows the vulvar superficial anatomy. 

## 4. Exam Methodology

Before starting the examination, verbal informed consent should be requested from the patient about the benefits and limitations of this exam. It is also recommended to explain that in elderly patients with atrophy or eroded lesions, this technique could cause discomfort. The patient should be placed in a gynecological chair in the lithotomy position in a comfortable and warm environment that guarantees privacy. A surgical glove or probe cover should be applied to protect the probe, and after use, the probe should be cleaned with antiseptics. After applying the probe cover, a generous amount of gel should be spread on the probe, which should be placed perpendicular to the skin without exerting excessive pressure to avoid distorting the image. Vulvar sonographic scan should be performed following a standard methodology (Figure 3):The vulva should be virtually visualized as a clock face and divided into four quadrants according to the ISSVD nomenclature [14] by an imaginary vertical line that passes through the clitoris and the anus (defining the lateral sides—right and left) and a horizontal line from the upper border of the hymenal ring (defining the anterior and posterior portions) (Figure 3A). This allows the vulva to be studied in its entirety and to describe the location and extension of vulvar lesions accurately.The probe should be placed in a transverse position at the beginning of the examination, and the clitoral area should be assessed first. From here, the examination should continue clockwise to study the remaining areas (Figure 3B).Once the clockwise rotation is completed and the clitoral area is reached again, the probe should be placed longitudinally and, starting at the level of the vestibule, moved laterally, first one way and then the other, extending it to the outer edge of the labia majora (Figure 3C). During this phase, an additional amount of gel should be applied to increase the distance between labia minora and majora to distinguish the different layers better [4]. A linear probe, preferably > 15 mHz (HFUS), is recommended to achieve optimal skin layer stratification [15]. The physiological presence of pubic hairs can alter the sonographic image. For this reason, some authors indicated hair removal the day before the imaging is performed.

## 5. Vulvar Lesion Description

The ultrasound protocol for the description of a suspected vulvar lesion should start with a 2D gray scale evaluation, describing:Localization (using the ISSVD nomenclature: anterior/posterior portion, right/left side, and central/lateral among right or left side) [14].Dimensions (measuring the lesion in the three axes longitudinally using a linear probe aligned with the major axis of the lesion) and transverse scan (linear probe aligned with the minor axis of the lesion).Volume (in cm^3^, estimated according to the ellipsoid formula: D1 (cm) × D2 (cm) × D3 (cm) × 0.52).Type of growth (exophytic or flat).Shape (oval, round or jagged).Appearance of the edges (regular or irregular; well-defined or non-well-defined).Depth of invasion of the skin layers [9].Echogenicity, which depends on the intrinsic capability of the tissues for reflecting the sound waves. The main echogenicity derives from the main components of the structure, which could be fluid, collagen, fatty tissue, or calcium. The sonographer should also consider the possibility of artifacts. A fluid-filled lesion will show anechoic (mostly black) or hypoechoic (gray) echogenicity and could have a posterior acoustic enhancement. A calcified mass will present a hyperechoic pattern and a posterior acoustic shadowing artifact [16].

At the end of this step, a comparison of the morphology with the perilesional and contralateral sites is recommended [16]. Subsequently, using Color or Power Doppler, the identification and measurement of vascularization (distribution, color score 1–4, type, peak systolic velocity of the afferent arteriole) should be performed [15]. Color Doppler should be set with a valuable threshold for detecting blood flow and a velocity ≥ 2 cm/s [16]. 3D reconstruction is optional but may be helpful for the clinician to understand the shape of the lesion and how it relates to surrounding structures [16]. Finally, if a lesion with sonographic features suspicious of malignancy is found, an evaluation of the inguinal lymph node stations according to the consensus opinion from the Vulvar International Tumor Analysis should be carefully performed [17].

## 6. Normal Vulvar Appearance

The sonographic appearance of the normal vulva reflects the different histology of its structures and, thus, their embryogenetic origin. Therefore, the knowledge of the differences in the skin of the various vulvar structures can help the sonographer navigate through the anatomical vulvar sites [6,15].

### 6.1. Epidermis

The epidermis consists of a multistratified squamous epithelium with different degrees of keratinization. Its thickness differs between different areas of the vulva due to the different histological features, age, and hormonal status. Migda et al. found that epithelial thickness at the level of the labia majora and the labia minora ranges from 0.21–0.15 mm to 0.08 mm, respectively, while mons pubis shows a mean epidermal thickness of 0.12–0.13 mm [6]. Additionally, pre-menopausal women may present an up to 40% thicker epidermis compared to post-menopausal patients [6]. However, measuring the epidermal layer is easily subject to error, as it is very thin to identify. Some authors recommended carrying out multiple thickness samplings and averaging them to limit the error. Automatic calculation software has been developed to overcome operator-dependent errors [6].

Mons pubis, labia majora, and the perineal body are ectoderm-derived structures and, therefore, are made up of keratinized stratified squamous epithelium, with the presence of sebaceous sweat glands and hairs [10]. At ultrasound scan, their epidermis appears as a continuous hyperechogenic layer resulting from the interface between the gel and the epidermis surface [6,18]. 

The labia minora are of mesodermal origin and are made up of poorly keratinized stratified squamous epithelium and are devoid of glands and hair follicles [4].

In areas in which the stratum corneum is particularly thick, the epidermis may be visible as a double hyperechoic “binary” line [4,15].

The vulvar vestibule and the urethral meatus are derived from the endodermis and, therefore, present nonkeratinized, generally highly glycogenated, stratified, and squamous epithelium [10]. In our experience, the lack of stratum corneum makes the epidermis of these areas poorly echogenic.

### 6.2. Dermis

The dermis shows a hyperechoic ultrasound appearance, even less bright than the epidermis. In its context, there may be hyperechoic reflections of collagen fibers and hypoechoic spots originating from the extracellular matrix between the collagen fibers [5]. Because the amount of dermal collagen depends mainly on the age of the patient, echogenicity and dermal thickness may vary in the population [15]. However, Migdal et al. found that dermis thickness is broadly similar in all vulvar structures, reaching 2.20–2.21 mm in the labia majora, 1.93 mm in the labia minora, and 1.78–1.82 mm in the pubic mons, without differences between pre- and post-menopausal women [6]. Dermis thickness in various body regions changes with age and appears to be correlated with BMI. While this correlation has been demonstrated for some anatomical areas, such as the abdomen and arms, it has not yet been confirmed in other areas, such as the vulva. Future studies correlating dermal thickness of the female external genital area and other factors, such as BMI, age, hormonal status, and therapies, are needed [19]. Hair follicles, sebaceous glands, and blood vessels are mainly present in structures of ectodermal origin (mons pubis and labia majora) and are optimally visualized at HFUS [6]. Sometimes, when HFUS is performed, a low-echogenic band can be found between the epidermis and dermis, called the “Subepidermal Low-echogenic band” (SLEB), related to water retention in the papillary dermis. This finding is most often associated with dermatitis (i.e., eczema or psoriasis), exposition to ultraviolet radiation, lichen sclerosus, and advanced age, while it is not detectable when hairs are removed [4,5].

### 6.3. Hypodermis

The hypodermis consists mainly of fatty lobules, and it is represented by a hypoechoic layer, interrupted by some hyperechoic filaments called fibrous septa [15].

Figure 4 shows the appearance of a normal vulva in pre-menopausal women.

### 6.4. Clitoris

The clitoris, being a complex organ, is a mixture of different histologies. Because of its length and three-dimensional development, it cannot be displayed on a single scan plane. Therefore, for a complete anatomical study of the clitoris, scans should be obtained in all three dimensions: transverse, coronal, and sagittal. Figure 5 represents a schematization of the visualization of the clitoris in three dimensions. Starting with a cross-section, the corpora cavernosa can be detected on the upper part of the vulva, lying on the pubic symphysis. They originate from the mesoderm and are visualized as two solid round structures with well-defined, hypoechogenic, symmetrical margins; cross-sections of cavernous arteries can be spotted in the center, preferably by color Doppler [20]. In the most cranial portion, the corpora cavernosa fuse medially in the raphe, visualized as a hypoechogenic line containing the clitoral artery. Anterior to the corpora cavernosa, the glans, derived from the mesoderm, can be found at the superficial level. It comprises relatively dense fibroconnective tissue with small, intercalated blood vessels. On cross-section, it appears as a hypoechogenic round structure with multiple vascular lacunae. The prepuce, placed on the top of the glands, is of ectodermal origin. Therefore, it is composed of keratinized multi-stratified squamous epithelium, and its epidermis should be visualized as a hyperechogenic layer [20,21]. By tilting the probe, the coronal plane can be obtained, showing, below the glans, the two corpora cavernosa and the two clitoral bulbs, which form the vaginal vault. The clitoral bulbs are of mesodermal origin and have a similar echogenicity as the corpora cavernosa, except that they have poorly defined margins because they are not covered by the albuginea tonaca [20]. Finally, on the sagittal plane, the clitoral bulbs and corpora cavernosa can be observed in their length, and the angle they form with the glans (clitoral angle) can be measured [20]. Figure 6 and Figure 7 provide cross-sections of sonographic images of post-menopausal (Figure 6) and pre-menopausal patients (Figure 7).

## 7. Abnormal Vulvar Ultrasound Appearance

### 7.1. Benign Vulvar Lesions

Cystic dilatation of obstructed Bartolini ducts is the most common benign finding in the vulva. They appear as fluctuant structures in the posterior third of the labia majora and contain clear mucous fluid, so their primary echogenicity is anechoic to hypoechoic. On HFUS, these lesions are located in the hypodermis (while the epidermis and dermis are intact) and have well-defined, avascular, anechoic borders. Echo enhancement is often seen posterior to the cyst due to enhancement through transmission of the cyst contents. If intracystic infection occurs, septa and sediment may be detectable, with suffused borders due to the inflammatory reaction [15].

### 7.2. Vulvar Dermatosis and Precancerous Lesions

Lichen sclerosus is a skin dermatosis characterized by chronic inflammation associated with itching, mainly diagnosed and monitored by vulvoscopy. However, it is reported that transperineal HFUS may help evaluate vulvar lichen due to its ability to better delineate the borders and, therefore, to study the treatment response. Zhou et al. compared the histological findings of lichen sclerosus, characterized by epidermal atrophy, follicular plug, collagen homogenization in the superficial dermis, and lichenoid lymphocytic infiltration with the ultrasound features [8]. Interestingly, the SLEB feature was found in 100% of cases, and its thickness seemed to reflect the collagen homogenization, while the hypoechoic dermal band had a heterogeneous appearance on ultrasound in cases of lymphocytic infiltration [8]. There is a lack of literature on the application of HFUS in assessing vulvar intraepithelial neoplasia (VIN). In a report on vulvar dermatoses, high-grade VIN was described as a circumscribed nodular lesion limited to the epidermis layer, and SLEB was also highlighted [22]. Figure 8 contains sonographic images of a vulvar lichen sclerosus in a post-menopausal patient, while Figure 9 shows its comparison with a normal vulva in a young woman.

### 7.3. Malignant Vulvar Lesions

Few studies have addressed the sonographic appearance of vulvar malignant lesions. Dermatological studies have been published on the appearance of cutaneous squamous cell carcinomas (SCCs) at HFUS, which most frequently appear as hypoechogenic or heterogeneous solid lesions with irregular margins and a high degree of vascularization. They usually do not present hyperechoic spots. Several authors reported that the vertical thickness of SCC could represent an important prognostic factor of disease progression. Therefore, its measurement is essential, especially in cutaneous SCC, where the skin layers are thin (i.e., in the vulvar labia minora), and there is a higher chance of infiltrating deeper layers [16]. In a case series conducted in 2019, HFUS was reported in the pre-operative assessment of vulvar carcinoma margins. Of the three patients with vulvar SCC enrolled, one had an advanced infiltrating lesion for which margin assessment was not feasible. In the other two patients, it was possible to study the depth of invasion and the extension of the ultrasound margins, which were revealed to be wider than the macroscopic evaluation and local palpation, demonstrating the need for larger excision margins [9].

The ultrasound technique for estimating the depth of ultrasound invasion of vulvar tumors should follow a standardized approach based on the European Society of Gynaecological Oncology’s 2023 [23] guidance for reporting the histologic degree of invasion. The ESGO guidelines suggest two different methods:Measurement from the adjacent most superficial dermal papilla to the deepest point of invasionMeasurement from the basement membrane of the deepest adjacent dysplastic (tumor-free) rete ridge to the deepest point of invasion, which should be the technique of choice. Because epidermal ridges cannot be distinguished on ultrasonography, our suggestion is to take the lower hyperechogenic line of the epidermal layer as a reference (Figure 10).

A recent interesting application of HFUS is the assessment of Extramammary Paget Disease (EMPD), an intraepidermal adenocarcinoma developing in areas rich in apocrine glands [24,25]. It usually appears as multiple, ill-defined erythematous plaques associated with itching and lichenification. At HFUS, the lesion presented a homogeneous hypoechogenic creeping area in the epidermis due to neoplastic cells typical of this disease with large nuclei and prominent nucleoli in small groups and single cells. EMPD may also show increased vascularization (usually associated with an invasive growth pattern) and a hyperechogenic layer on the epidermal surface due to abnormal keratinization with heterogeneous posterior acoustic shadowing. Although these features are nonspecific for the diagnosis of EMPD, which remains a histological diagnosis, they could guide the extent and invasion degrees of the lesion compared to perilesional healthy skin. Moreover, several studies have been proposed to measure the depth of lesion infiltration to identify cases of EMPD with an invasive growth pattern. It was found that the vertical infiltrate thickness of the lesion correlates with an invasive form when using cut-off ranges from 1.55 cm to 1.65 cm (with a positive predictive value of 62% and 44%, respectively). It is important to remember that inflammation can cause an overestimation of vertical thickness because the inflammatory infiltrate in the dermis cannot usually be distinguished sonographically from a tumor infiltrate [26].

In our experience, almost all malignant vulvar lesions present a solid consistency. The most frequently encountered echogenicity is hypoechogenic, with a predominantly non-homogeneous pattern. Many lesions show irregular margins and intense vascularization (color score 3–4 at the CD). Figure 11, Figure 12, Figure 13 and Figure 14 show four cases of malignant vulvar lesions examined by an experienced sonographer in our hospital before surgery.

## 8. Potential Applications

Vulvar ultrasound may be helpful in different aspects of vulvar pathology. In case of swellings such as serous cysts, lipomas, abscesses, and hemangiomas, a vulvar ultrasound can examine the content of the lesion and guide its management. For lichen sclerosus, it has been proposed as an additional parameter to the clinical and vulvoscopic evaluation and to assess the response to medical treatment. In managing patients affected by vulvar malignancies, vulvar ultrasound may play a critical role in different aspects. Regarding local staging, it could define the tumor dimensions and the depth of stromal invasion, providing a better distinction between stage IA and stage IB. The distinction between stage IA (tumor size ≤ 2 cm and stromal invasion ≤ 1 mm) and stage IB (tumor size > 2 cm or stromal invasion > 1 mm) is crucial [28] because systemic staging and groin treatment are needed for stage IB [29]. The sonographic evaluation of the stromal invasion may be beneficial in bigger lesions, where the incisional biopsy may not be representative of the maximum stromal invasion.

Moreover, the actual dimensions of a vulvar lesion may not be adequately assessed only by physical examination, mainly when the growth is toward the deep vulvar planes. Vulvar ultrasound may also aid in more objectively measuring the distance of the lesion from the midline (decision between unilateral or bilateral groin assessment) [29,30] and to assess the eligibility for sentinel lymph node procedure (unifocal tumor < 4 cm) [29]. Local staging may also be improved by vulvar ultrasound by evaluating the extension to the urethra, vagina, or anus [28,29]. In fragile patients with multiple comorbidities who cannot tolerate longer or more invasive diagnostic tests, vulvar ultrasound may improve the primary lesion definition and, thus, the treatment planning [30]. The comprehensive sonographic evaluation of vulvar malignant lesions may also allow the modulation of surgical excision and predict the need for reconstruction by flap surgery. 

## 9. Conclusions

Vulvar ultrasound is a rapidly spreading technique that has demonstrated a good ability to distinguish between the different vulvar structures and correctly measure and map the vulvar lesions. It has excellent potential in distinguishing between benign and malignant lesions, local staging of vulvar malignancies, estimating the depth of invasion, and guiding the surgeon at the time of excision. Its use also appears promising in the follow-up of patients undergoing non-surgical therapies to evaluate the response to treatment. Vulvar ultrasound is an economical test that can be performed in every hospital with minimal equipment; it is minimally invasive and, in case of patient discomfort, can be performed pre-operatively under narcosis. On the other hand, highly skilled operators are required to correctly distinguish anatomical structures despite the distortion caused by the disease. Furthermore, in particularly large lesions, the mass cannot be wholly included in a single scan, and the deeper margins are difficult to include and investigate by ultrasound. In the future, further studies will be necessary to evaluate both the extension of the margins and the degree of invasiveness more precisely in the tumor in the subepidermal skin layers to provide the patient with the most suitable surgical approach possible.

## Figures and Tables

**Figure 1 diagnostics-15-00627-f001:**
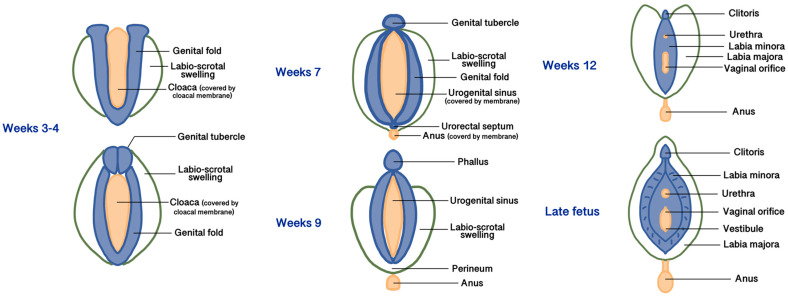
Main phases of embryogenesis of female external genitalia.

**Figure 2 diagnostics-15-00627-f002:**
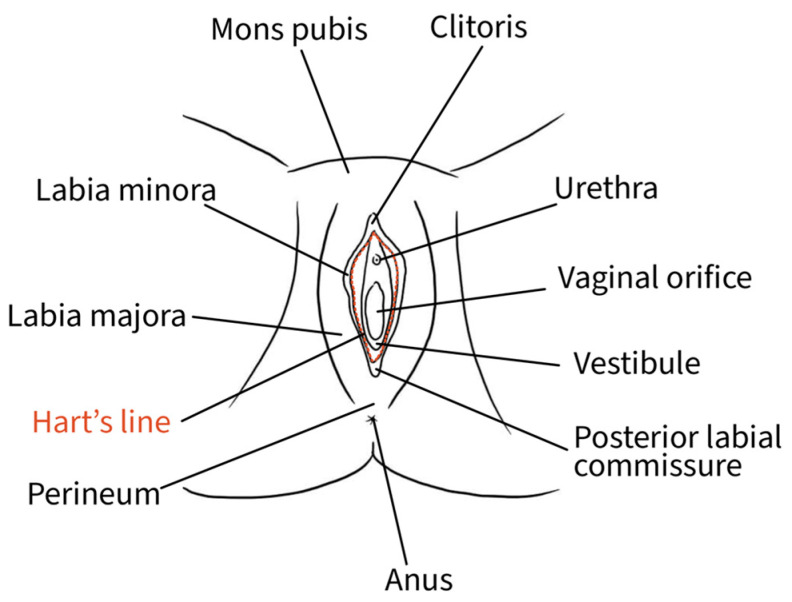
Schematic representation of superficial vulvar structures.

**Figure 3 diagnostics-15-00627-f003:**
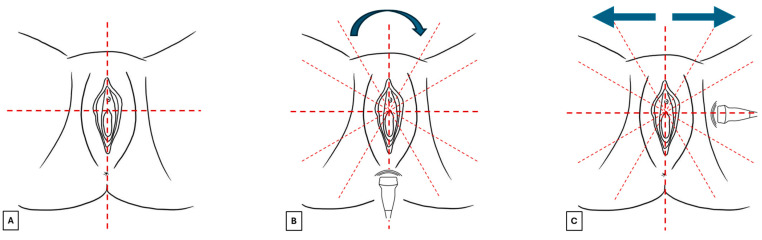
Images (**A**–**C**) summarize the standard scan methodology to perform a vulvar ultrasound.

**Figure 4 diagnostics-15-00627-f004:**
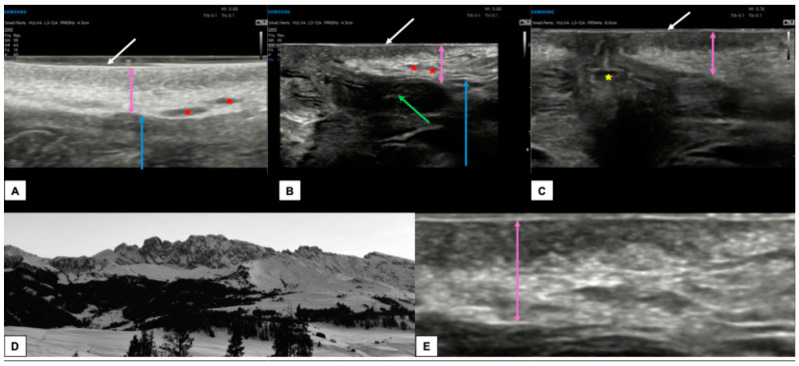
Ultrasound appearance of a normal vulvar in a 32-year-old woman (**A**–**C**,**E**) with hair removal. In all three images, the white arrow represents the epidermis, visible as a double hyperechoic “binary” line. Immediately below, the dermis (purple arrows in (**A**–**C**,**E**)) is visible as a layer composed of a hypoechogenic serrated area superiorly and a hyperechogenic area inferiorly. The appearance of the dermis in young women with normal vulva resembles the outline of a “Mountain Ridge” (**D**,**E**). The thickness of both epidermis and dermis can vary between the sites of the scan ((**A**): perineal area, (**B**): labia minora and majora, (**C**): perianal area). The green arrow lies on the left labia minora fold (**B**), the blue arrow is placed on the hypodermis (**A**,**B**), the red asterisk represents the hair follicles (**A**,**B**), while the yellow asterisk represents the anal introitus (**C**).

**Figure 5 diagnostics-15-00627-f005:**
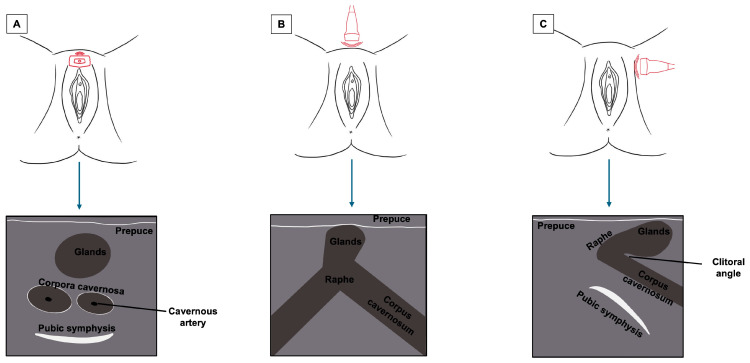
Summary of the sonographic visualization of the clitoris in three dimensions ((**A**) cross-section plane; (**B**) coronal plane; (**C**) sagittal plane).

**Figure 6 diagnostics-15-00627-f006:**
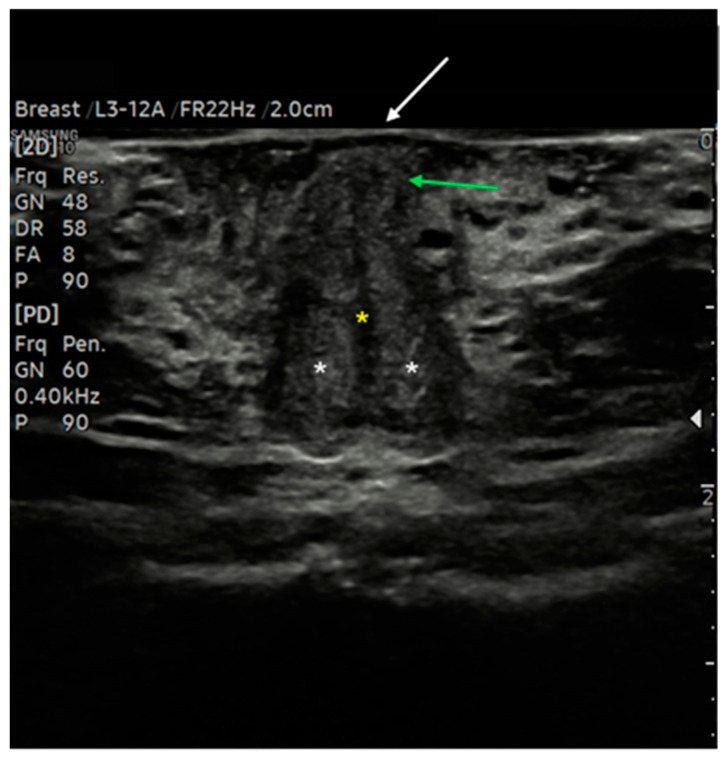
Ultrasound cross-section of the clitoris in a post-menopausal woman. Corpora cavernosa (white asterisks) merge medially into the raphe, which appears as a linear hypo-anechoic structure (yellow asterisk). Anterior to the raphe is visualized the initial part of the glans (green arrows), covered by the prepuce, with the epidermis appearing as a thick hyperechoic line (white arrow).

**Figure 7 diagnostics-15-00627-f007:**
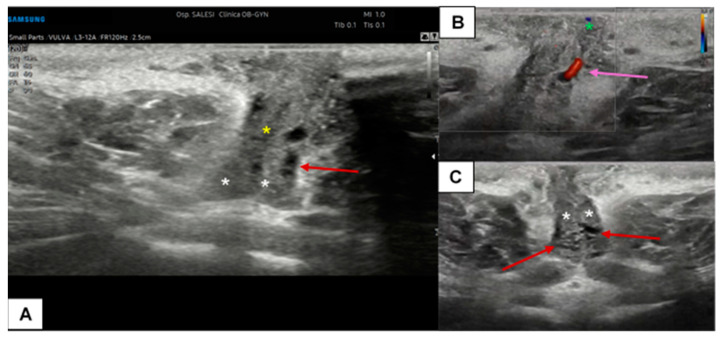
Ultrasound cross-section of the clitoris in a pre-menopausal woman with clitoralgia (**A**–**C**). Corpora cavernosa (white asterisks), the raphe (yellow asterisk), and the glands (green asterisk in (**B**)) are clearly detectable. With the support of CD assessment, the left cavernous artery was easily identified (purple arrow in (**B**)). In this patient, the vascular plexus of the corpora cavernosa was particularly dilated (visible in (**A**,**B**), red arrows), probably due to an inflammatory event.

**Figure 8 diagnostics-15-00627-f008:**
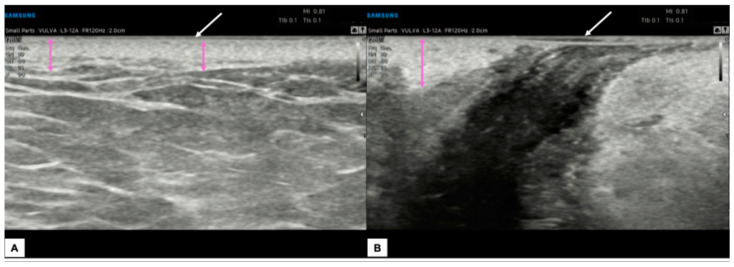
Ultrasound appearance of a vulva affected by lichen sclerosus in the perianal and peri-introital right vulva area, histologically confirmed. In both figures (**A**,**B**), the epidermal layer is visible (white arrows), represented by two parallel hyperechogenic lines. The dermal layer (purple arrows) maintains a variable thickness according to the vulvar anatomical zones and appears particularly dense and hyperechogenic without the hypoechogenic serrated area superiorly. The most striking feature is the presence of thick, hyperechogenic fibrous septa that cross the hypodermis into the dermis.

**Figure 9 diagnostics-15-00627-f009:**
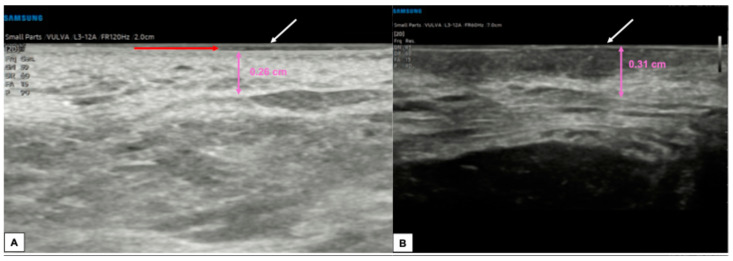
Comparison between the sonographic appearance of the cutaneous layers of a vulva affected by lichen sclerosus in a post-menopausal patient (**A**) and a normal vulva in a young woman (**B**). The shape, thickness, and echogenicity of the epidermal layer (white arrows in (**A**,**B**)) are essentially overlapping. Skin dermis thickness (purple arrows in (**A**,**B**)) is similar in the two cases, with loss of the upper hypoechogenic serrated dermal area and increased echogenicity in the dermis affected by lichen sclerosus (**A**), probably due to all those processes leading to collagen homogenization. Interestingly, we found in our case of lichen sclerosus (**B**) the presence of SLEB (red arrow in (**A**)), already described by other authors [4,5,8], which is typical of dermatoses and present in 100% of lichen sclerosus cases, according to Zhou et al. [8]. The SLEB can be visualized as a well-defined and homogeneous hypoechogenic band located between the epidermis and the dermis and could represent water retention in the papillary dermis.

**Figure 10 diagnostics-15-00627-f010:**
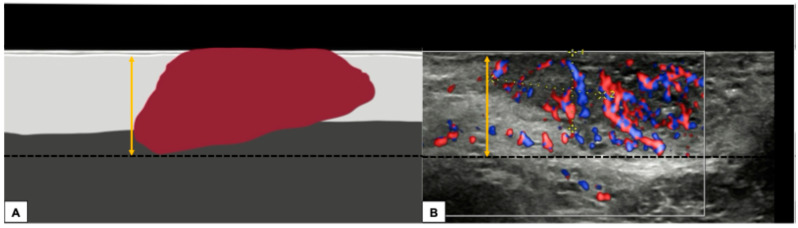
Graphical representation of a standardized methodology for calculating the depth of invasion (**A**) and a practical example on a sonographic image of a vulvar cutaneous cancer (**B**), according to technique number 2 by ESGO guidelines. The black dashed line demarcates the deepest point of invasion of the tumor, while the double yellow arrow estimates its distance from the basement membrane of the lower hyperechogenic epidermal line and corresponds to the depth of histological invasion.

**Figure 11 diagnostics-15-00627-f011:**
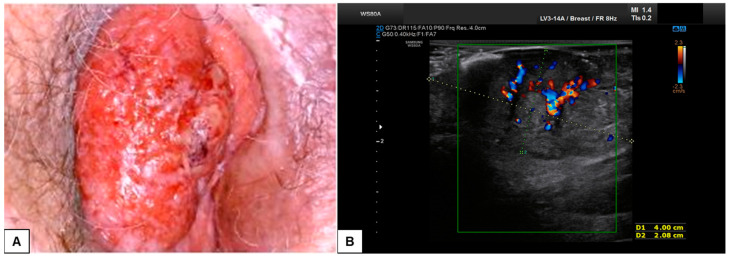
Patient 1: comparison between vulvoscopy (**A**) and ultrasound appearance (**B**) of the same vulvar lesion. The tumoral mass involved the right labia minora and the paraclitoral region. At sonographic scan (**B**), the lesion appeared solid, hypo-anechoic, with regular and well-defined borders, and measured (antero-posterior extension × depth × latero-lateral extension) 4.5 × 2.1 × 3.5 cm. Hart’s line was not clearly detectable, probably because of dermal tumoral infiltration. At the CD assessment, the mass was highly vascularized (Color Score 3–4), with double central vessels feeding the tumor. After surgical excision, on histological examination, the lesion was defined as a moderately differentiated SCC, with 0.5 cm depth of invasion and TNM staging pT1b N0 (FIGO staging Ib).

**Figure 12 diagnostics-15-00627-f012:**
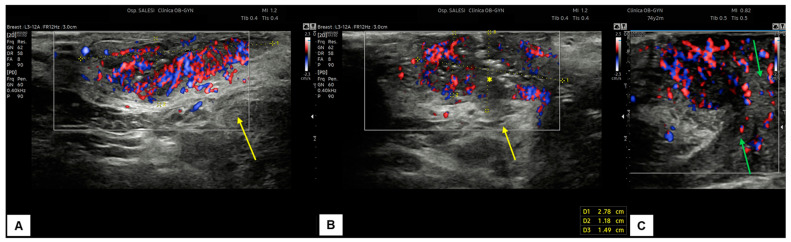
Patient 2: ultrasound scan of a vulvar lesion (**A**,**B**) mainly involving the right periclitoral area. At the sonographic examination, the mass appeared solid, with inhomogeneous hypoechogenic echogenicity and irregular borders, measuring 2.7 (antero-posterior extension) × 1.5 (depth) × 2.0 (latero-lateral extension) cm. Some hyperechogenic spots were spotted inside the mass (yellow asterisk in (**B**)), corresponding to the areas of ulceration. The lesion clearly involved the dermis layer (yellow arrows in (**A**) and (**B**)), the right corpora cavernosa, and the glans (green arrows in **C**). An intense vascularization pattern (Color Score 3–4) was demonstrated at CD assessment (**A**–**C**). After a radical vulvectomy followed by Y-flap reconstruction, histological examination revealed a poorly differentiated ulcerated SCC, with high cytological grading (G3) and an invasion depth of 0.5 cm. Final staging pT1bpN0 (FIGO Ib).

**Figure 13 diagnostics-15-00627-f013:**
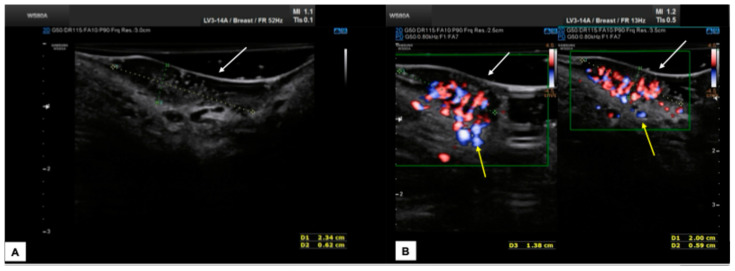
Patient 3: sonographic scan of a vulvar malignancy characterized by multiple cerebroid lesions in a 36-year-old patient. The primary tumor involved the right hemivulva, the perineal area bilaterally area, and the left labia minora. The ultrasound appearance of perineal lesions is shown in (**A**,**B**). Both in (**A**,**B**), the keratinized perineal epidermis is clearly detectable as a single hyperechogenic layer at ultrasound (white arrows). The tumors were elevated compared to surrounding skin, with a solid appearance, hypoechoic inhomogeneous pattern, well-defined and irregular margins, and intense vascularization (Color Score 4 at the CD). The widest lesion measured 2.3 (antero-posterior extension) × 0.6 (depth) × 1.8 (latero-lateral extension) cm. The interruption of the dermal layer and therefore dermal invasion was demonstrable in several scans (yellow arrows in (**B**)). The patient underwent radical excision of the right hemivulva, left labia minora, and perineum. In all samples, the histological result was infiltrating keratinizing SCC, with a minimum invasion depth of 1 mm up to 5 mm and free excision margins of 1.0–2.0 mm. Final staging pT1bN0 (FIGO Ib). Subsequently, the patient underwent a widening of the excision of the right hemivulva and left perineal area [27]. The second histological examination revealed no malignancy on the surgical specimens, and the postoperative follow-up was regular. The patient finally decided to undergo HPV vaccination.

**Figure 14 diagnostics-15-00627-f014:**
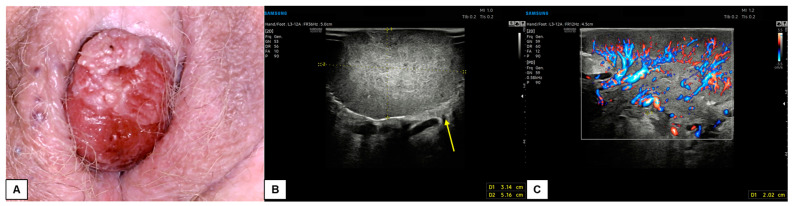
Patient 4: comparison of vulvoscopic visual (**A**) and vulvar ultrasonography (**B**,**C**) of a bleeding vulvar malignancy with exophytic development and cerebroid shape. The primary tumor involved the right hemivulva and the clitoral area. The mass had a solid, hypoechoic inhomogeneous pattern at ultrasound with well-defined and irregular margins. The lesion diameters measured 5.2 (antero-posterior extension) × 3.1 (depth) × 5.1 (latero-lateral extension) cm. Intense vascularization was visualized at CD (Color Score 3–4), with a 2.0 cm-thick vascular pedicle from the right hemivulva. Dermal invasion was demonstrable in several scans (yellow arrow in (**B**)). The patient underwent a simple vulvectomy. The histological result revealed an ulcerated, well-differentiated (G1), HPV-independent infiltrating SCC, with an invasion depth of 3 mm. Final staging pT1bNx (FIGO Ib).

**Table 1 diagnostics-15-00627-t001:** Embryonic layers and precursors from which vulvar structures originate.

Embryonic Layer	Precursor	Vulvar Anatomical Parts
**Ectoderm**	Labioscrotal swelling	**Labia majora**
	Labioscrotal swelling	**Mons pubis**
	Labioscrotal swelling	**Perineum**
**Mesoderm**	Genital fold	**Labia minora**
**Endoderm**	Cloaca	**Anus**
	Urogenital sinus	**Urethra**
	Urogenital sinus	**Vestibule**
**Mixed origin**	Genital tubercle	**Clitoris**

## Data Availability

Not applicable.

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
