# Peer review of "Transperineal Vulvar Ultrasound: A Review of Normal and Abnormal Findings with a Proposed Standardized Methodology"

_diagnostics, 2025, doi:10.3390/diagnostics15050627_

Round 1
Reviewer 1 Report
Comments and Suggestions for Authors
Dear authors, I read you article with interest. Here are some my suggestions.
Figure 8: white arrows on the left part of B - are they correct?
line 473: 1.8 (latero-lateral extension) cm - 1.3?
I would suggest adding at least 3 pictures:
- normal vulva - US showing layers from epidermis down
- how is stromal invasion measured
- lichen - characteristics on US
Also, do you have any data or experience with vaginal probe? Is this something you would consider including in the article?
Author Response
Dear reviewer, thank you very much for your valuable suggestions and comments on our manuscript.
Comment 1: Figure 8: white arrows on the left part of B - are they correct?
Response: Thank you for pointing this out. We corrected the white arrows on that image (after the revision, now is the number 13, picture B)
Comment 2: line 473: 1.8 (latero-lateral extension) cm - 1.3?
Response: Thank you for this comment. We checked the image again, and we confirm that the widest lesion's latero-lateral extension was 1.8 cm (not showed in the images)
Comment 3: I would suggest adding at least 3 pictures:
- normal vulva - US showing layers from epidermis down
- how is stromal invasion measured
- lichen - characteristics on US
Response: Thank you for your suggestion. We have included additional images to our manuscript (figure 4: normal vulvar appearance; figures 8 and 9: lichen sclerosus appearance and comparison with normal vulva; figure 10: standardized methodology for measuring the degree of stromal infiltration). About the definition of infiltration depth, we have inserted lines 438-448 to explain a standardized method for measuring it.
Comment 4: Also, do you have any data or experience with vaginal probe? Is this something you would consider including in the article?
Response: Thank you for your interest. The linear probe, at the present time, appears to be the best instrument both because of its good image resolution and because of its shape, which is perfectly suited to the study of skin surfaces. Endovaginal probes usually work at a frequency ranging from 2 to 11 MHz, which is insufficient to have a good visualization of the vulvar skin layers. In addition, the endovaginal probe has a narrow tissue contact area, and is not sufficient to explore the extent of vulvar lesions. However, transvaginal ultrasound can be used in large vulvar tumors to evaluate cranial invasion of other pelvic organs, such as the vagina, urethra, bladder and rectum.
Reviewer 2 Report
Comments and Suggestions for Authors
This manuscript represents a well-written description of transperineal vulvar ultrasound. Its use of graphic anatomical illustrations, along with sonographic images, perfectly conveys the diagnostic value that US has for supplementing our current diagnostic tools for assessing vulvar pathology.
This reviewer has nothing to offer for recommending any revisions to it, as it should be published in its current form, to contribute to the current medical literature on this subject.
Author Response
Thank you very much for your response.
We are grateful that you appreciated our work.
Best regards